# The leishmaniases in Kenya: A scoping review

**Grace Grifferty**[1,2]*, **Hugh Shirley**[2,3], **Katherine O'Brien**[2,4], **Jason L. Hirsch**[5], **Adrienne M. Orriols**[2,6‡], **Kiira Lani Amechi**[7‡], **Joshua Lo**[8‡], **Neeharika Chanda**[1‡], **Sarra El Hamzaoui**[4‡], **Jorja Kahn**[9‡], **Samantha V. Yap**[10‡], **Kyleigh E. Watson**[4‡], **Christina Curran**[11‡], **Amina Atef AbdelAlim**[10‡], **Neeloy Bose**[12‡], **Alissa Link Cilfone**[13], **Richard Wamai**[2,14,15,16]

**1** Department of Cellular and Molecular Biology, College of Science, Northeastern University, Boston, Massachusetts, United States of America, **2** African Centre for Community Investment in Health, Nginyang, Baringo County, Kenya, **3** Harvard Medical School, Boston, Massachusetts, United States of America, **4** Department of Health Sciences, Bouvé College of Health Sciences, Northeastern University, Boston, Massachusetts, United States of America, **5** The Ohio State University College of Medicine, Columbus, Ohio, United States of America, **6** University of Florida College of Medicine, Gainesville, Florida, United States of America, **7** Department of International Affairs, College of Social Sciences and Humanities, Northeastern University, Boston, Massachusetts, United States of America, **8** Department of Mathematics and Department of Psychology, College of Science, Northeastern University, Boston, Massachusetts, United States of America, **9** Department of Behavioral Neuroscience, College of Science, Northeastern University, Boston, Massachusetts, United States of America, **10** Department of Biology, College of Science, Northeastern University, Boston, Massachusetts, United States of America, **11** Department of Biochemistry, College of Science, Northeastern University, Boston, Massachusetts, United States of America, **12** Department of Bioengineering, College of Engineering, Northeastern University, Boston, Massachusetts, United States of America, **13** Northeastern University Library, Northeastern University, Boston, Massachusetts, United States of America, **14** Department of Cultures, Societies and Global Studies, College of Social Sciences and Humanities, Integrated Initiative for Global Health, Northeastern University, Boston, Massachusetts, United States of America, **15** Department of Global and Public Health, University of Nairobi, Nairobi, Kenya, **16** Nigerian Institute of Medical Research, Federal Ministry of Health, Lagos, Nigeria

☯ These authors contributed equally to this work.
‡ AMO, KLA, and JL also contributed equally to this work. NC, SEH, JK, SVP, KEW, CC, AAA, and NB also contributed equally to this work.
* gracegrifferty@gmail.com

## Abstract

### Background

The leishmaniases are a group of four vector-borne neglected tropical diseases caused by 20 species of protozoan parasites of the genus *Leishmania* and transmitted through a bite of infected female phlebotomine sandflies. Endemic in over 100 countries, the four types of leishmaniasis–visceral leishmaniasis (VL) (known as kala-azar), cutaneous leishmaniasis (CL), mucocutaneous leishmaniasis (MCL), and post-kala-azar dermal leishmaniasis (PKDL)–put 1.6 billion people at risk. In Kenya, the extent of leishmaniasis research has not yet been systematically described. This knowledge is instrumental in identifying existing research gaps and designing appropriate interventions for diagnosis, treatment, and elimination.

### Methodology/Principal findings

This study used the Preferred Reporting Items for Systematic Reviews and Meta-Analyses (PRISMA) methodology to determine the state of leishmaniases research in Kenya

**Data Availability Statement:** All relevant data are within the manuscript and its Supporting Information files.

**Funding:** The author(s) received no specific funding for this work.

**Competing interests:** The authors declare no competing interests.

and identify research gaps. We searched seven online databases to identify articles published until January 2022 covering VL, CL, MCL, and/or PKDL in Kenya. A total of 7,486 articles were found, of which 479 underwent full-text screening, and 269 met our eligibility criteria. Most articles covered VL only (n = 141, 52%), were published between 1980 and 1994 (n = 108, 39%), and focused on the theme of "vectors" (n = 92, 34%). The most prevalent study types were "epidemiological research" (n = 88, 33%) tied with "clinical research" (n = 88, 33%), then "basic science research" (n = 49, 18%) and "secondary research" (n = 44, 16%).

## Conclusion/Significance

While some studies still provide useful guidance today, most leishmaniasis research in Kenya needs to be updated and focused on prevention, co-infections, health systems/policy, and general topics, as these themes combined comprised less than 4% of published articles. Our findings also indicate minimal research on MCL (n = 1, <1%) and PKDL (n = 2, 1%). We urge researchers to renew and expand their focus on these neglected diseases in Kenya.

## Author summary

The leishmaniases are a group of four vector-borne neglected tropical diseases (NTDs) that are endemic in over 100 countries, putting over 1.6 billion people at risk. In Kenya, the extent of research on leishmaniasis remains unclear. Therefore, this scoping review aims to uncover and classify the body of literature on leishmaniasis in Kenya to elucidate gaps in knowledge and inform future research, related health policies, and interventions in Kenya. Through a database search, we identified 269 articles that met our eligibility criteria, which were included in our final analysis. Our analysis revealed that the majority of articles discussed VL. Most articles were published between 1980 and 1994. There was uneven distribution between the themes of published articles, with vector-related research dominating the list, followed by treatment, diagnostics, general epidemiology, and pathophysiology. The most prevalent type of study was a tie between epidemiological and clinical research, followed by basic science research and then secondary research. There is minimal research coverage on MCL and PKDL, which is as expected given the low prevalence of these diseases in Kenya. Furthermore, little research spans prevention, co-infections, health systems/policy, and general topics.

## Introduction

The leishmaniases are a group of neglected tropical diseases (NTDs) caused by intracellular parasites of the *Leishmania* genus and targeted for elimination globally by 2030 [1]. The parasite is spread by *Phlebotomus* and *Lutzomyia* sandflies between humans and from reservoir species, such as the rock hyrax, to humans [2]. The distinct entities within this group are visceral (VL), cutaneous (CL) and its subtypes, localized and diffuse CL, and mucocutaneous leishmaniasis (MCL). A fourth manifestation, a post-treatment sequela of VL called post-kala-azar dermal leishmaniasis (PKDL), rounds out the clinical manifestations of this parasitic infection [3]. While many infected individuals may be asymptomatic, the outcomes for clinical

leishmaniasis range from resolution to disfiguring scars from CL, PKDL, and MCL to death from untreated VL [4]. VL in Kenya is more frequently caused by certain species, such as *L. donovani*, while CL is seen with *L. tropica* [3, 5]. Therefore, the relative frequencies of each manifestation depend on the prevalence of the various parasite species in each country and region.

Across Southeast Asia, the Middle East, Africa, and South America, approximately 1.6 billion people are at risk of infection [6]. The World Health Organization (WHO) estimates that 50,000–90,000 people contract VL and 600,000–1 million contract CL annually [7]. Estimates of the burden of MCL and PKDL are unavailable. PKDL plays an essential role in maintaining the disease within a population between epidemic periods, as latent infection can be maintained for years before becoming clinically apparent with infectious lesions. Known risk factors for infection include poverty, pastoral living, and sleeping outdoors or near livestock [8].

The clinical diagnoses and treatment of VL, CL, MCL, and PKDL are complicated by non-specific presentations early in the course of each disease [9]. Co-infection with HIV presents an important manifestation of leishmaniasis infection as its clinical presentation is often atypical and, therefore, further obscures accurate diagnosis in communities with high HIV prevalence [10]. Tuberculosis (TB) co-infection can complicate treatment further. Studies have estimated that the prevalence of TB in HIV-VL co-infected individuals may be between 5.7% and 29.7% [11]. TB infection leaves the host more susceptible to *Leishmania* infection, creating a co-infection (often TB-VL) that can lead to poor prognosis [12, 13]. Because the *Leishmania* parasite and the tuberculosis bacterium employ similar models of macrophage inhabitation, and thus trigger similar immune responses from the host, co-infection is challenging to diagnose and often remains undiscovered for years [13].

The gold standard of VL diagnosis involves invasive spleen aspirates while screening in the field includes direct agglutination testing (DAT) or rK39 antigen-based testing [12, 14]. CL can be diagnosed with skin scrapings, fine-needle aspirations, or punch biopsies of the ulcerated skin or mucosal lesions, allowing for direct visualization of *Leishmania* parasites [14]. Polymerase chain reaction (PCR) and parasite culture are also used [14]. MCL is often diagnosed with PCR, as the mucosal lesions of MCL are frequently devoid of parasites [14]. Diagnosing PKDL is complex because clinical presentation can vary by geographic location but it generally presents as a rash [14]. Serological or molecular tests are used to confirm PKDL diagnosis [15].

Accurate diagnosis of leishmaniasis is vital, as current treatment modalities are not without significant side effects. Treatment regimens for VL in East Africa include pentavalent antimonials such as sodium stibogluconate plus paromomycin for 17 days. Second line therapies include formulations of amphotericin B or miltefosine [12]. CL and MCL, while not life-threatening, can leave disfiguring scars that impact quality of life [12]. Though pharmacotherapy that include topical paromomycin or pentavalent antimonial intralesional injections and thermo- and cryotherapy have demonstrated efficacy and are recommended [12], the common approach for CL in Kenya is intralesional injections. While the use of oral miltefosine tablets to treat CL and VL is approved for South America, it is not yet widely available in East Africa with the first study conducted in Ethiopia for CL and for VL in Kenya showing promise [16–18]. PKDL is often treated with sodium stibogluconate in East Africa, while the use of miltefosine has been reported elsewhere [19]. Nevertheless, treatment for the leishmaniases can be prohibitively expensive, and providers often rely on the donation of supplies and medications to care for their patients [8].

Kenya, a burgeoning country in Sub-Saharan Africa, is among five East African countries that now bear the largest proportion of the global burden of VL [20]. The disease was first

described in Kenya in 1935 in the northern districts of Mandera and Wajir [21]. Since that time, outbreaks of both VL and CL have occurred in various parts of the country [22]. However, the exact status of leishmaniasis endemicity in Kenya is not well understood [22]. There is inadequate documentation on the prevalence, burden, and spatial distribution of the leishmaniases [22]. According to one estimate, there are approximately 1,600 cases of VL annually, resulting in the loss of over 13,000 DALYs and over 170 deaths [23]. The combined incidence of MCL and CL is estimated at over 580 annually, resulting in a loss of 253 DALYs [23]. VL is endemic in arid and semi-arid regions in the Rift Valley, Eastern region, and Northeastern regions [22]. The most critical transmission foci are currently Baringo, Isiolo, Marsabit, West Pokot, Turkana, Kitui, Garissa, and Wajir counties [22]. In addition, CL has been reported in the Mt. Elgon region, the Rift Valley and Nakuru and Nyandarua counties with a recent outbreak investigation in Nakuru county identifying a potentially large burden [22, 24]. MCL and PKDL data is scarcer than VL and CL data in Kenya. It is estimated that PKDL occurs in less than 5% of VL cases in Kenya, although research on PKDL is not as common [9]. It is within this context that leishmaniasis research has been conducted in Kenya for the last 90 years. This scoping review examines the body of literature on research conducted on leishmaniasis in Kenya, recognizes gaps, and provides guidance on future research direction. This is critical, especially because Kenya recently launched a national strategy (2021–2025) to accelerate control and eventually eliminate leishmaniasis [22].

## Methods

We followed the Preferred Reporting Items for Systematic Reviews and Meta-Analysis (PRISMA) guidance for scoping reviews (S1 Checklist). The protocol was registered in Open Science Framework on May 14, 2022 at https://doi.org/10.17605/OSF.IO/5GJW7.

### Study identification

A comprehensive query of PubMed, Web of Science, Embase, ClinicalTrials.gov, Cochrane CENTRAL, WHO ICTRP, and the Pan African Clinical Trials Registry was conducted to identify relevant articles. Searches for each database were crafted by a librarian (ALC) using a combination of controlled vocabulary and natural language terms. Each search was informed by and tested against a selection of 'gold standard' articles shared by team members (RW, JK, SM). Once crafted, team members (RW, JK, SM) shared input on additional search terms, and the searches were subsequently peer-reviewed by another librarian (PEC) with expertise in systematic review searching. Each search was created with the intent to capture the broadest set of possible results related to Kenya and leishmaniasis within that database. Search strategies for PubMed, Embase, Web of Science, and Cochrane CENTRAL are available in S1 Search Strategy. Searches in all databases were initially conducted between October 22 and 24, 2019. Before being uploaded to Rayyan, a tool to facilitate independent article screening, results were deduplicated in an EndNote library [25].

The search was restricted to English language articles published any time before and including October 2019. After screening for the results originally identified was nearly finished, a second update search was run in all the same databases on January 25, 2022 to capture search results published between October 2019 and January 25, 2022. These results were deduplicated using the same method described above before being uploaded into Rayyan for screening. Records identified and screened are detailed in Fig 1.

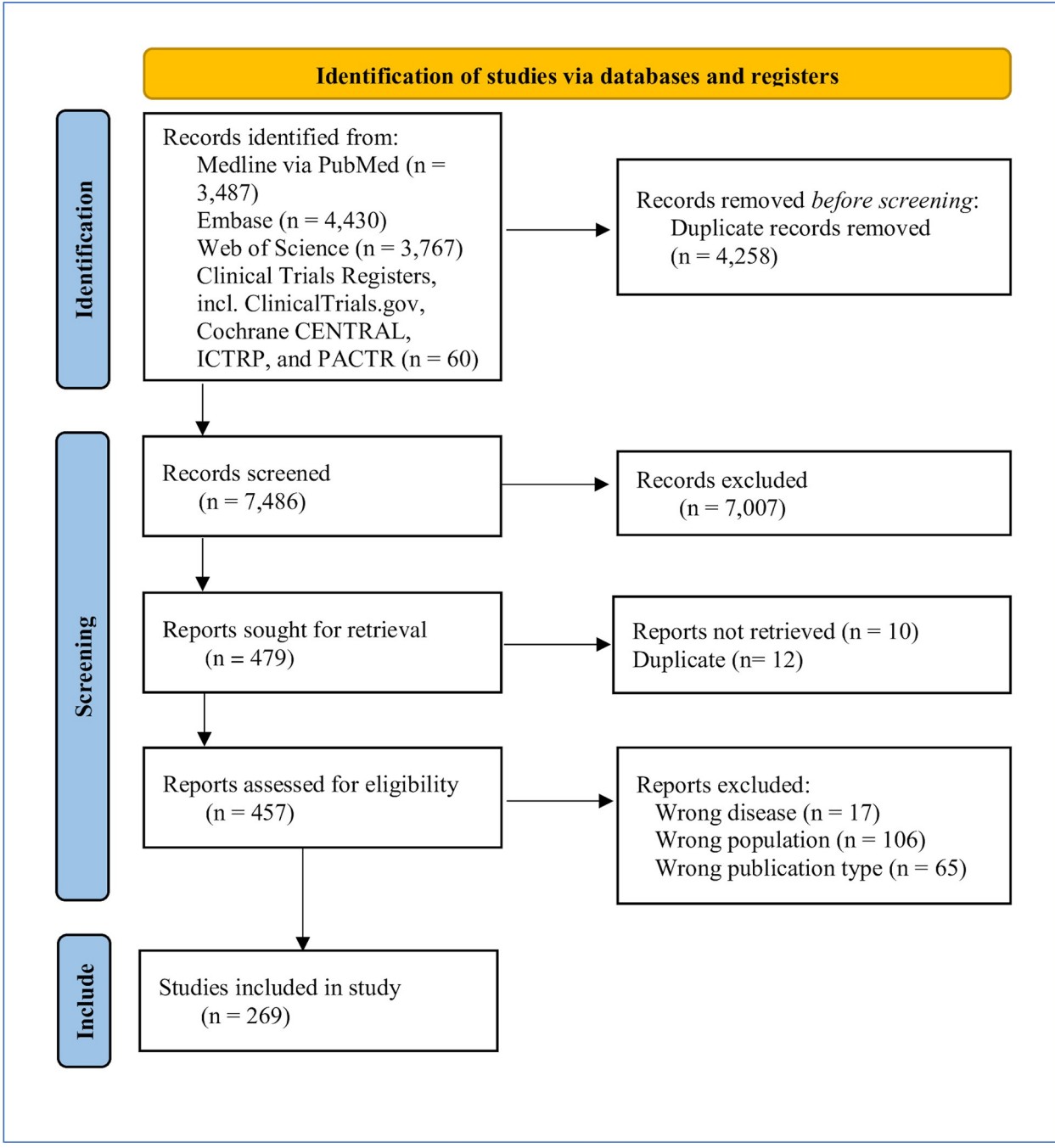

**Fig 1. PRISMA 2020 flow diagram including searches of databases and registers only** [26].

## Study selection

**Title and abstract screening.** All articles were independently screened in Rayyan based on title and abstract, or only title if the abstract was not available, by ten assessors (GG, KO, JK, SM, CH, JH, JL, NC, KA, SEH) that formed pairs of reviewers, ensuring that all articles were double screened against the inclusion and exclusion criteria (Fig 2). If a document did

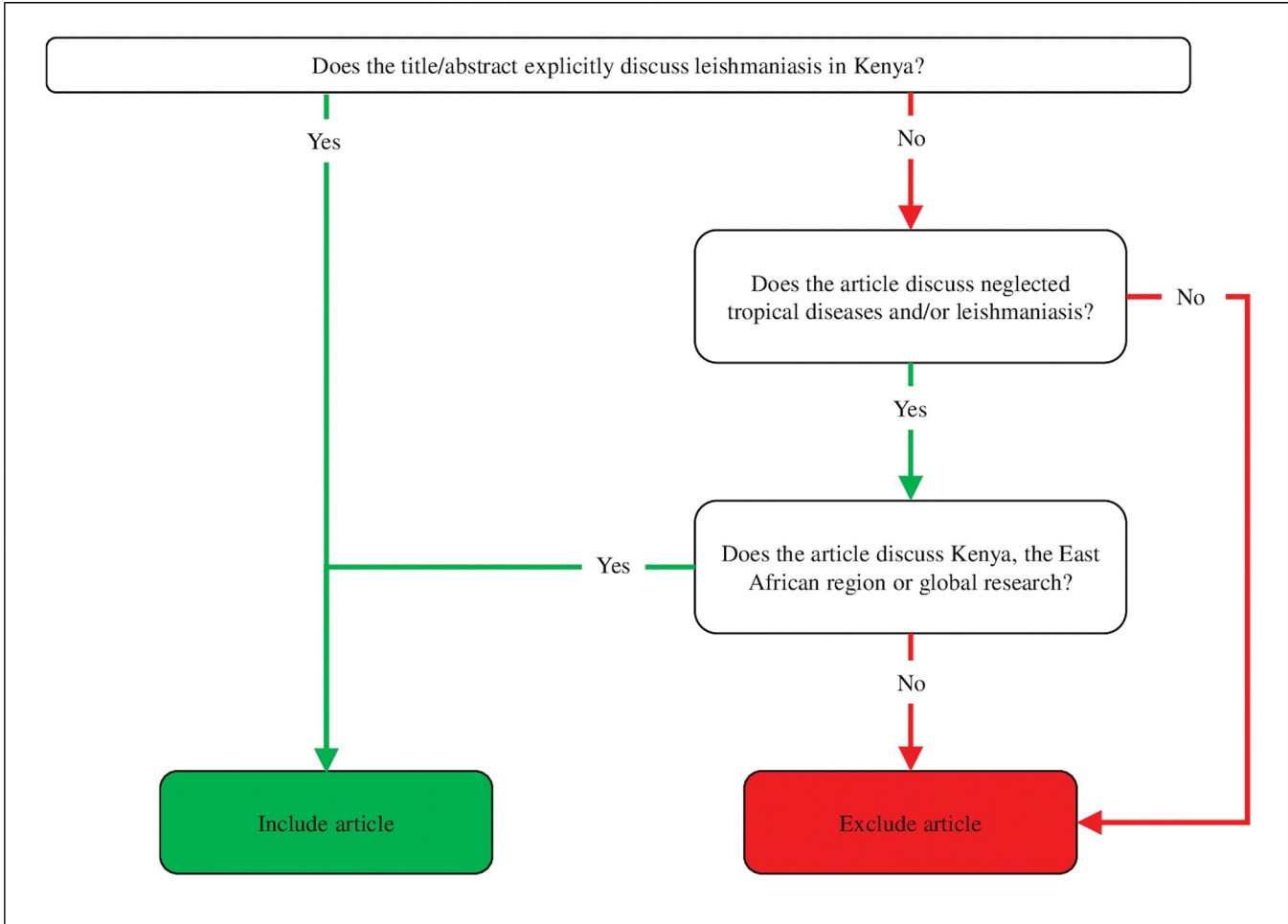

**Fig 2. Inclusion and exclusion criteria for title and abstract screen.**

not explicitly meet the inclusion criteria, an inclusive approach was taken, and it was considered for full-text screening. GG or KO resolved conflicts. See the outcome of this step in Fig 1.

## Full-text screening

Given the size of the project, we imposed additional limits on the inclusion criteria to make the project scope more feasible. Therefore, we chose only to include peer-reviewed journal articles with traditional structures (i.e., introduction, methods section, results, conclusion/discussion) and institutional reports. Grey literature was excluded. As a result, our exclusion category titled "wrong publication type" included the following publication formats: abstract (n = 2), book (n = 1), conference abstract (n = 25), correspondence (n = 10), laboratory meeting notes (n = 1), news flash (n = 2), presentation (n = 1), seminar (n = 1), short communication (n = 19), unpublished clinical trial (n = 2). One exception was made for the publication *Tropical and Geographical Medicine*, which is a book of peer-reviewed articles.

A full-text screen was conducted by 12 assessors in Rayyan (GG, KO, HS, JH, JL, KA, AO, SY, KW, CC, AA, NB) that formed pairs of reviewers, ensuring that all full-texts were

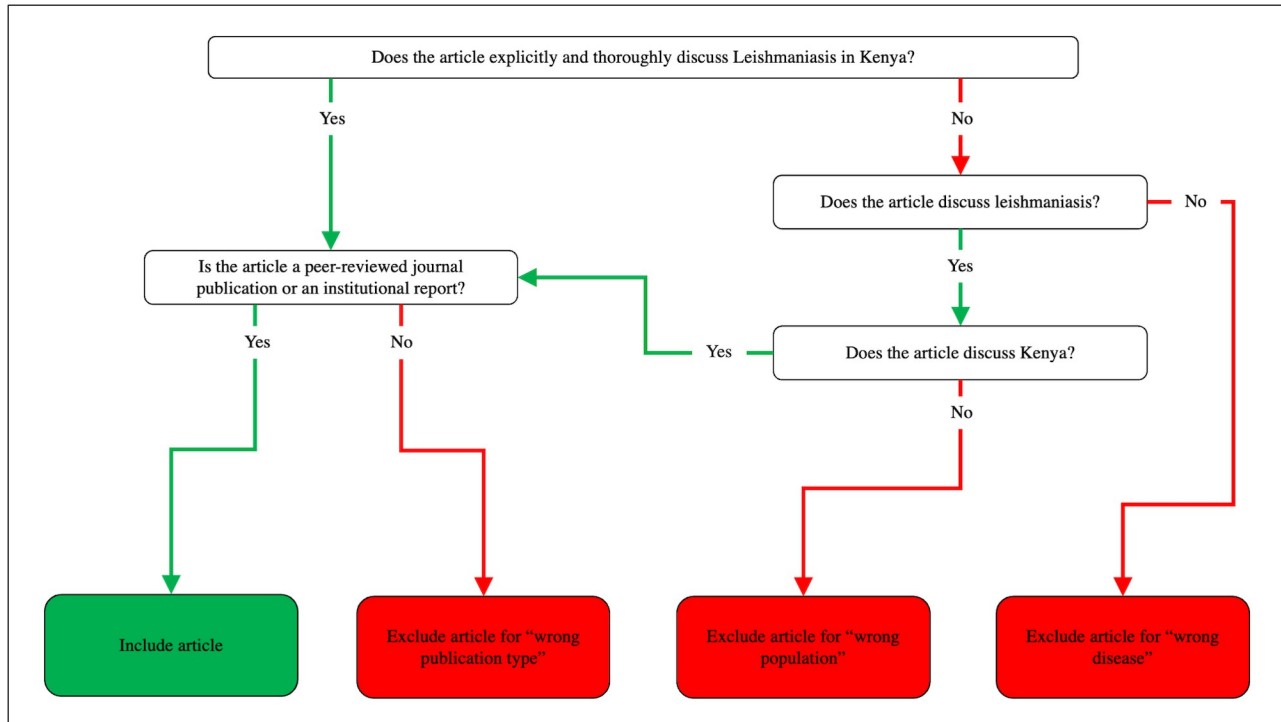

**Fig 3. Inclusion and exclusion criteria for the full-text screen.**

double screened against the inclusion and exclusion criteria (Fig 3). Themes and subthemes were developed to facilitate categorization of papers into a framework that would allow quantitative comparisons between study design and focus over time. "Theme" as defined by S1 Table and summarized in Table 1 were also added to each article by each reviewer in Rayyan. Conflicts regarding inclusion and exclusion criteria, as well as the theme designation, were resolved by GG or KO. Only articles that explicitly and thoroughly discussed leishmaniasis in Kenya were included for data extraction at this stage. See the outcome of this step in Fig 1.

## Data extraction

All data in Rayyan was exported into Microsoft Excel, either automatically or manually. Objective data, including year of publication, funding source, and number of times cited, was manually extracted from each article (GG, HS, KO and AO) and added to the Microsoft Excel database. A codebook was developed by GG, HS, and KO (S1 Table) to further classify included articles by type of leishmaniasis (i.e., forms of clinical presentation of leishmaniasis), type of study, and subcategory of study. Due to the complicated nature of these labels, a pilot study was conducted between GG, HS, KO, and AO to refine these labels in the codebook to resolve any potential discrepancies between reviewers and increase inter-reviewer reliability. Each article was read in full by two of four reviewers (GG, HS, KO, AO), and the type of leishmaniasis, type of study, and subcategory of study labels were added to the Microsoft Excel database. A third reviewer resolved conflicts. A summary of all manually added data labels can be found in Table 1. Individual included articles and their labels are available in S1 Data.

**Table 1. Summary of manually added labels.**

| Type of extracted data | Labels | Subcategory (if applicable) |
|---|---|---|
| **Type of leishmaniasis** | VL | - |
| | CL | - |
| | MCL | - |
| | PKDL | - |
| | VL/CL | - |
| | VL/MCL | - |
| | VL/PKDL | - |
| | CL/MCL | - |
| | CL/PKDL | - |
| | MCL/PKDL | - |
| | Multiple | - |
| | Not specified | - |
| **Theme** | General epidemiology | - |
| | Prevention | - |
| | Pathophysiology | - |
| | Diagnostics | - |
| | Treatment | - |
| | Health systems/policy | - |
| | Vectors | - |
| | Co-infections | - |
| | General topics | - |
| **Type of study** | Basic science research | Animal study |
| | | Cell study |
| | | Genomics/ Genetics/ Computational |
| | | Biochemistry |
| | | Method development |
| | | Other |
| | Clinical research | Clinical study |
| | | Diagnostic study |
| | | Prognostic study |
| | | Case report/series/study |
| | | Other |
| | Epidemiological research | Molecular epidemiology |
| | | Method development |
| | | Cohort study |
| | | Case-control |
| | | Cross-sectional |
| | | Monitoring/surveillance |
| | | Other |
| | Secondary research | Meta-analysis |
| | | Systematic review |
| | | Scoping review |
| | | Literature/narrative review |
| | | Bibliometric review |
| | | Policy studies |
| | | Opinion/viewpoint |
| | | Institutional report |

(*Continued*)

**Table 1.** (Continued)

| Type of extracted data | Labels | Subcategory (if applicable) |
|---|---|---|
| Year of publication | - | - |
| Funding source | - | - |
| Number of times cited | - | - |

## Results

### General results

After title/abstract screening and full-text screening, 269 articles were included for final analysis. There was a slow increase in the number of publications from 1945–1979, and then an uptick between 1980 and 1994 (n = 108, 39%) followed by another uptick from 2010–2014 (n = 36, 13%) (Fig 4). The majority of articles covered VL (n = 141, 52%), followed by CL (n = 50, 19%), and then articles that covered both VL and CL (n = 37, 14%). Only six articles covered VL and PKDL together (2%), while two articles covered only PKDL (1%) (Fig 5). Only one article covered MCL (<1%) (Fig 5). There is minimal research coverage of MCL and PKDL independently, and articles covering multiple types of leishmaniasis. The most prevalent themes (definitions available in S1 Table) were vectors (n = 92, 34%), treatment (n = 51, 19%), a tie between diagnostics (n = 40, 15%) and epidemiology (n = 40, 15%), followed by pathophysiology (n = 36, 13%) (Fig 6). There is a clear lack of research on prevention, co-infections, health systems policy, and general topics as these themes combined comprised less than 4% of published articles (Fig 6). The types of study categorizations were more evenly distributed, with the most prevalent type of study being a tie between epidemiological (n = 88, 33%) and clinical research (n = 88, 33%), followed by basic science research (n = 49, 18%) and secondary research (n = 44, 16%) (Fig 7). Over half of all epidemiological research articles fell under the subcategory of monitoring/surveillance (n = 51, 58%). Almost a quarter fell under the subcategory of molecular epidemiology (n = 21, 24%) (Fig 7). Other subcategories together comprised only 18% of articles (Fig 7). Clinical research articles were more evenly distributed amongst

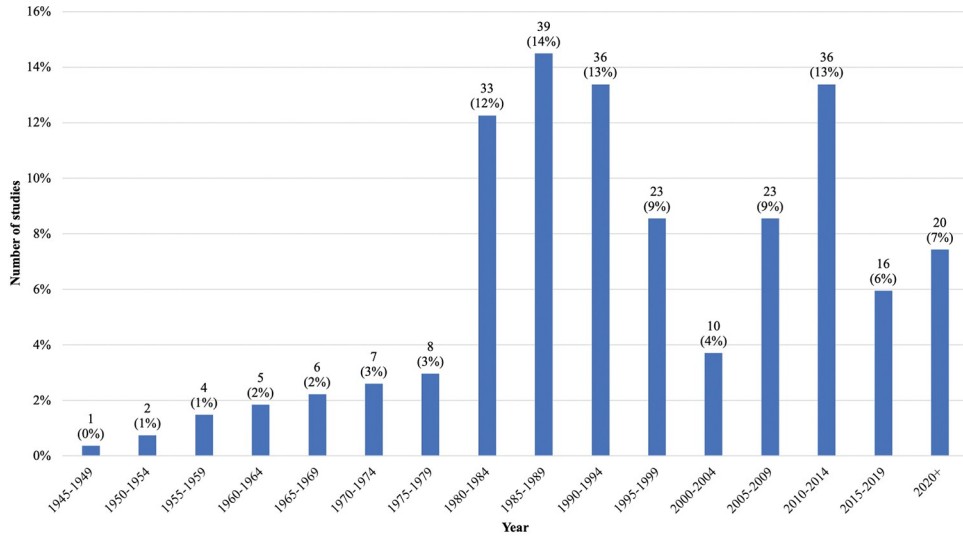

**Fig 4. Results of full-text screen: Years articles were published.**

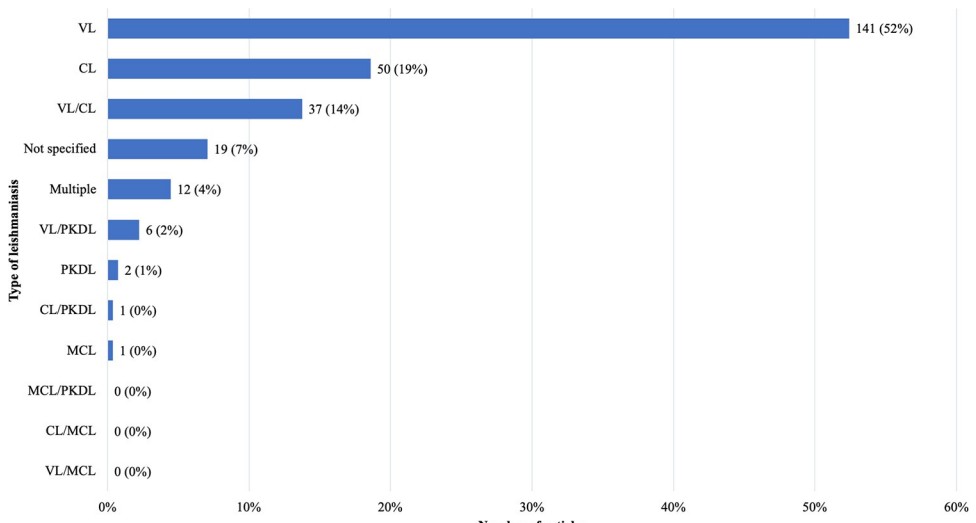

**Fig 5. Results of full-text screen: Number of articles per type of leishmaniasis.**

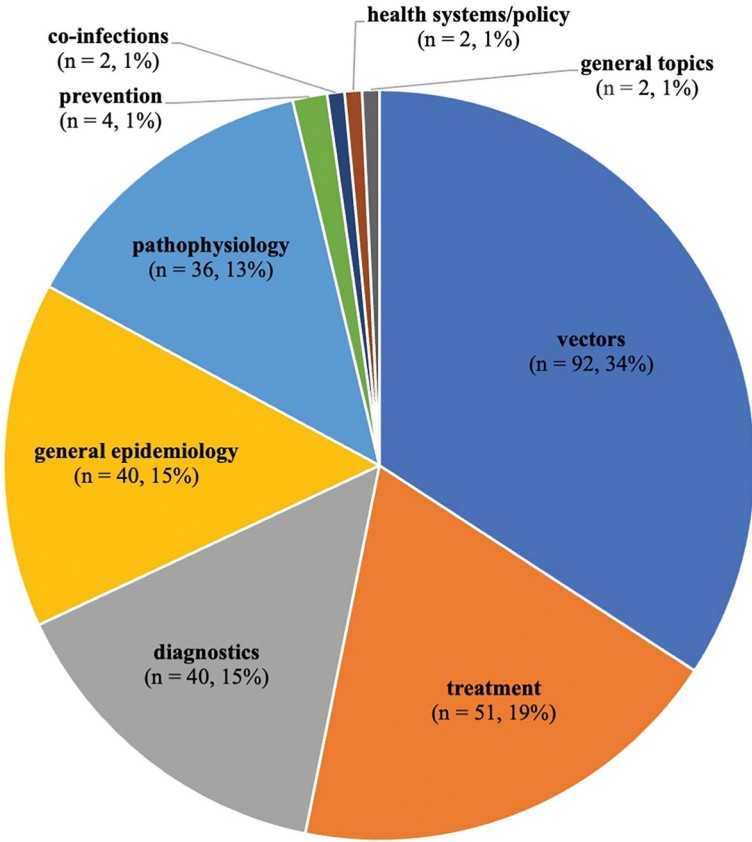

**Fig 6. Results of full-text screen: Number of articles per theme.**

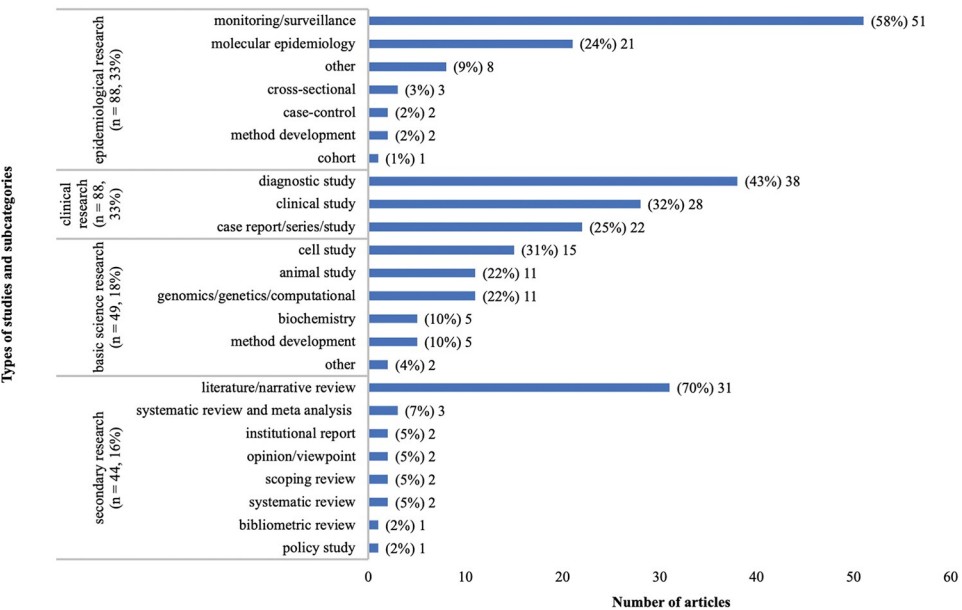

**Fig 7. Results of full-text screen: Number of articles per types of study and subcategory.**

subcategories, which included diagnostic studies (n = 38, 43%), clinical studies (n = 28, 32%) and case report/series/studies (n = 22, 25%) (Fig 7). Basic science research articles comprised mostly of cell studies (n = 15, 31%), followed by a tie between animal studies (n = 11, 22%) and genomics/genetics/computational studies (n = 11, 22%) (Fig 7). Biochemistry, method development articles, and other articles comprised of less than a quarter of the articles (n = 12, 24%) (Fig 7). Lastly, most secondary research fell under the subcategory of literature/narrative reviews (n = 31, 70%) with all other subcategories containing three or fewer articles (Fig 7). It is important to consider, however, that some types of studies naturally attract fewer papers, most notably those in the secondary research category including meta-analyses, systematic reviews, scoping reviews, literature/narrative reviews, bibliometric reviews, and others. This may be one reason why the secondary research category contains the fewest number of articles. To further evaluate the nature of the secondary research, S7 Table was created. S7 Table reveals that the majority of literature/narrative reviews covered the theme general epidemiology (n = 14, 45%), followed by treatment (n = 7, 22%) and vectors (n = 6, 19%). It is important to note that there was no secondary research found focusing solely on prevention.

## Results by decade published

Additional gaps in research coverage were uncovered by accounting for the decades that articles in Figs 4–7 were published (S2–S4 Tables). S2 Table revealed that 30% (n = 42) of VL research was published in the 1980s and that 74% (n = 105) of VL research is at least 12 years old. CL research and VL/CL combined research predominantly occurred in the 1980s (n = 12, 24% and n = 12, 32%, respectively) and 1990s (n = 12, 24% and n = 12, 32%, respectively), with 74% and 76% of research being over 10 years old, respectively. Articles classified with the other types of leishmaniasis categories were sporadically published throughout the decades.

S3 Table revealed that epidemiological research peaked in the 1980s (n = 26, 30%) and 1990s (n = 32%), followed by an abrupt drop in publications since 2000. Clinical research went from a relatively large number of publications in the 1980s (n = 28, 32%) and 1990s (n = 19,

22%) to a drop in publications from 2000–2009 (n = 8, 9%), and then an uptick in the 2010s (n = 20, 23%). Basic science research was relatively evenly distributed throughout the decades since the 1980s, while secondary research was scarce up until the 2010s when it peaked (n = 17, 39%).

When analyzing themes published by decade, S4 Table showed a clear dominance of vector-focused articles (n = 92, 34%). Over 50% of articles with the themes of vector, diagnostics and pathophysiology were published in the 1980s and 1990s, rendering most research with those assigned themes over 20 years old. Treatment focused articles peaked in the 2010s (n = 18, 35%). Only the theme of general epidemiology has been evenly distributed over the last 70 years. Articles with the remaining themes were sporadically published over the last few decades.

### Funders and publishers of research

The top 10 funders of leishmaniasis research in Kenya are outlined in S5 Table, with a joint effort between United Nations Development Programme/World Bank/World Health Organization Special Programme for Research and Training in Tropical Diseases topping the list with 43 studies funded. In addition, we compiled data on the journals with the most publications (S1 Chart). No single journal overwhelming dominated research on leishmaniasis in Kenya. However, two journals together are responsible for 26% of all publications: *Transactions of the Royal Society of Tropical Medicine and Hygiene* (n = 36, 13%) with articles in the 1940s-1960s and 1980s-2010s, and the *East African Medical Journal* (n = 34, 13%) with articles from the 1970s to the 2010s. The top 10 most cited articles were found using Google Scholar and are identified in S6 Table. The majority of these studies covered VL (n = 6) and the majority were secondary research articles (n = 6), specifically literature/narrative reviews (n = 5). The most cited article dominated the list as it was cited 5,111 times. The second most cited article was cited 1,240 times.

### Discussion

To the best of our knowledge, this is the first comprehensive review of research on the leishmaniases in Kenya. Seventeen years ago, Tonui had published a situational narrative review of leishmaniases [27]. Using the scoping review methodology our study systematically expands the state of knowledge providing updated information on the situation in Kenya. Our review may be used as a roadmap to develop evidence-based research program for better implementation of the national strategic plan for leishmaniasis 2021–2025 and beyond.

This scoping review is timely given that the target for leishmaniasis elimination is set as 2030, and the call for enhanced control and elimination of VL in East Africa following remarkable declines in Southeast Asia [1, 20]. In this context, a virtual meeting of the WHO in June 2022 for VL programs in Central and East Africa and South Asia, attended by RW, reiterated the need for evidence-driven interventions, which emphasizes the critical importance for our scoping review. Owing to these goals, as well as the more complex nature of the disease in Eastern Africa due to comorbidities, migration, conflicts and ecological changes, the state of research on the leishmaniases across the fields established in this review in all of the East African countries need to urgently be established [20]. In January 2023, the WHO's Stakeholders' meeting for the Development of a Strategic Plan for the Elimination of Visceral Leishmaniasis in East Africa (attended by RW and KO) determined that VL in East Africa is eliminable by 2030, which is in line with the NTDs 2030 roadmap. Key discussions focused on key needs in research including reviews precisely like what we have done. Kenya will be the first country to have conducted a systematic review of existing research on leishmaniases. Technical experts recognize the value for such studies to inform research prioritization for elimination targets.

The call for operational and implementation research for optimizing existing interventions in diagnosis, treatment, health systems and vector control can be validated for Kenya by the findings of our review [20]. This review will not only fill a major gap in Kenya, but also serve as a model for similar reviews in other countries in the region given the common strategy now under development.

For the past five years in Kenya, there has been a marked increase in leishmaniasis cases reported without definitively established causes [22, 28]. According to the District Health Information System (DHIS2) in Kenya, a total of 1,463 VL cases were reported in 2019; compared to only 607 cases reported in 2018, and slightly less than the Global Burden of Disease (GBD) 2019 estimates [22, 23]. In addition, outbreaks of VL and CL continue to be reported in endemic counties, and now new areas are experiencing an emergence of leishmaniasis [22]. According to the Kenyan Ministry of Health, "the accurate disease burden is unknown as information on the prevalence, burden and spatial distribution of the disease is inadequate. . ." [22]. Our scoping review echoes this sentiment as we identified that 87% of articles about leishmaniasis in Kenya are over seven years old.

In 2021, the Kenyan Ministry of Health put forward a strategic plan for 2021–2025 to accelerate the gains achieved in controlling leishmaniasis and halt its resurgence [22]. Its mission is to ensure access to prevention, diagnosis, treatment, and control of leishmaniasis in Kenya and aims to do so by reducing mortality, morbidity, and outbreaks [22]. The strategy acknowledges that the management of leishmaniasis is hindered by the lack of data, cost, and access [22]. Our report addresses the lack of data component by highlighting specific research areas that need greater attention. We identified little research pertaining to MCL, which is as expected given low prevalence of the associated vector in Kenya [7, 24, 29]. We also noted a limited number of PKDL studies. Since PKDL patients act as reservoirs of VL, with a prevalence of 2–5% in VL-treated individuals in Kenya, the identification of these cases is important to control VL spread and therefore, we encourage additional research aimed at detecting, treating, and monitoring these cases [30, 31]. PKDL incidence in Kenya is unknown although reported as suspected by some treatment centers. For this reason, it is recognized in the national strategic plan as of importance in controlling VL, and as such we recommend PKDL incidence studies be conducted [22].

Additionally, articles examining co-infections, prevention, health systems/policy, and general topics are lacking across all leishmaniases. Research is especially needed on significant VL co-infections like HIV/AIDS and their implications for accurate diagnosis, treatment, and elimination as recently noted in the WHO guideline for the treatment of VL in HIV co-infected patients in East Africa and South-East Asia [32]. VL-TB co-infection is also chronically under researched and goes un-mentioned in the 2021–2025 Kenya Strategic Plan for Control of Leishmaniasis [22]. There are various case studies published on VL-TB and CL-TB co-infection outside of East Africa, showing the complexities and success of treatment [33–35]. More robust, quality data is needed in East Africa given that TB presents a barrier to successful VL treatment. We also recommend that even more thoroughly covered themes like vectors, pathophysiology, and diagnostics be given greater attention because 50% of all articles with those themes were published in the 1980s and 1990s. We also recommend a focus on epidemiological research, where 61% of articles were published in the 1980s and 1990s. Only secondary research has had significant coverage in the last two decades. Even so, we recommend additional secondary research be undertaken, specifically systematic reviews, meta-analyses, scoping reviews, policy studies, institutional reports, and opinion/viewpoints, which are lacking across all types of leishmaniasis in Kenya. Operational and implementation research is also lacking and is critical in informing behavioral and cost-effective interventions. Lastly, the

COVID-19 pandemic likely inhibited research and operations, so we expect a decrease in publications from 2020–2022 [36].

It is important to note that there are gaps in research on a regional level not addressed by this scoping review that warrant additional attention. A team led by Philippe Guérin at the Infectious Diseases and Data Observatory group at the University of Oxford has proposed a research agenda for outstanding universal research gaps in VL that include methodological, clinical, PKDL, pharmacological, diagnostics, and other aspects [37]. Additionally, research on more effective diagnostics (e.g., for VL-HIV co-infected population) and treatments (e.g., Miltefosine) for East Africa conducted elsewhere in Southeast Asia are lacking [38–41]. There is also greater deficiency of systematic research on reservoir and vector ecology and vector control in East Africa owing to conflicts, instability and population movement in the border regions [8, 42, 43]. Further, in the decade of NTDs to 2030, implementation research for leishmaniasis elimination, mapping and vector control activities remain gaps especially in Kenya as well as the rest of East Africa [1, 22].

## Strengths

The main strength of our study is that we have identified the whole body of English language literature pertaining to leishmaniasis in Kenya to the best of our abilities. We undertook an extensive and comprehensive search strategy, with input from a multidisciplinary team, to capture the broadest set of results possible related to Kenya and leishmaniasis within databases. Another strength of our scoping review is the thorough approach taken during title/abstract and full-text screening, where two assessors reviewed each article to minimize bias. This increased the likelihood that articles were correctly included or excluded, and assigned appropriate labels. Lastly, a comprehensive codebook was developed and a pilot study was completed for the more complicated labels (type of leishmaniasis, type and subcategory of study) to increase inter-reviewer reliability.

## Limitations

We acknowledge that there are several limitations to our study. Firstly, we only included English-language articles, which excludes potentially relevant non-English publications. Another limitation is that some articles were excluded because the full texts were unavailable to us (n = 10), so we may have excluded potentially relevant articles. Additionally, we excluded articles that were not in the form of peer-reviewed published journal articles or institutional reports. In total, we excluded 65 articles under the exclusion criteria of "wrong publication type." While they may have published relevant data, they were not in the desired format, and thus, we may have excluded relevant research. We believe it is important to note why three clinical trials were excluded. One was not yet completed (excluded under "wrong publication type"), another was already included in the form of a published peer-reviewed journal article in our screen (excluded under "duplicate"), and the last did not have any related peer-reviewed journal publication (excluded under "wrong publication type").

Additionally, during our full-text screen, we excluded articles that did not explicitly and thoroughly cover leishmaniasis in Kenya. Ultimately, this determination of what constitutes thorough coverage is subjective. To minimize this issue, two reviewers analyzed each article and made determinations to include or exclude the article. If they disagreed, a third reviewer was brought in. When Kenya was minimally mentioned in an article about leishmaniasis, the article was excluded for "wrong population." When leishmaniasis was minimally mentioned in an article about Kenya, the article was excluded for "wrong disease." Our definition of

"minimally mentioned" was up to two mentions unless the article was a secondary research piece analyzing multiple regions or diseases and included leishmaniasis and Kenya.

The authors acknowledge that there was a degree of classification bias in this study. To limit classification bias from the onset, the authors determined the "themes" to be used prior to beginning the screen. After the full-text screen was complete, a pilot study was conducted to create a codebook (S1 Table) and further classify included articles by type of leishmaniasis, type of study, and subcategory of study. The authors felt this was necessary to provide a more comprehensive analysis of the collected articles. However, it was at this stage that classification bias was present as these classifications were determined after having read through the full texts in the prior stage. However, the classification we used enabled us to identify topics of research relevant to the policy on the elimination of VL in Kenya.

Another limitation is that under the type of study category "epidemiological research", we only labeled articles with the subcategories of cohort study, case-control, or cross-sectional if the article explicitly self-described its methodology as such. These articles may have also fallen under "monitoring/surveillance" or "molecular epidemiology," but the other subcategories were given priority.

Several articles did not specify the type of leishmaniasis they were studying. If articles did not explicitly state a type of leishmaniasis, *Leishmania* species was used as a proxy given species are clinically associated with VL, CL, MCL, and/or PKDL [3, 22, 44, 45]. As this knowledge is still evolving, future research may reveal new links between *Leishmania* species and produce alternative classifications.

Finally, this scoping review did not identify disseminated or localized cutaneous leishmaniasis as independent types of leishmaniasis, and instead used the general classification of CL to label relevant articles. By identifying examining these more specific types of cutaneous leishmaniasis, we would have been better able to identify specific gaps in CL coverage.

## Supporting information

**S1 Checklist. Preferred reporting items for systematic reviews and meta-analyses extension for scoping reviews (PRISMA-scr) checklist.**
(PDF)

**S1 Search Strategy. Search strategies of databases.**
(PDF)

**S1 Chart. Journals with most articles published about leishmaniasis in Kenya.**
(TIF)

**S1 Data. Individual included articles and their labels.**
(XLSX)

**S1 Table. Codebook for manual labels.**
(PDF)

**S2 Table. Articles published by decade per type of leishmaniasis.**
(TIF)

**S3 Table. Type of study by decade.**
(TIF)

**S4 Table. Themes of articles by decade.**
(TIF)

**S5 Table. Top 10 funders of leishmaniasis research in Kenya.**
(TIF)

**S6 Table. Top 10 most cited articles.**
(PDF)

**S7 Table. Subcategories of secondary research articles by theme.**
(TIF)

## Acknowledgments

The authors acknowledge the contributions of Sukanya Mittal and Connor Holmes at Northeastern University for their work in the title and abstract screening phase. We also thank Philip Espinola Coombs at the Northeastern University Library for his contributions to the methods development. Finally, we acknowledge Dr. Davis Wachira (deceased May 2022), who was the longtime focal point person for the national leishmaniasis program in Kenya's Ministry of Health, for his wisdom, humility, and dedicated service to leishmaniasis programs in Kenya.

## Author Contributions

**Conceptualization:** Grace Grifferty, Jorja Kahn, Richard Wamai.

**Data curation:** Grace Grifferty, Hugh Shirley, Katherine O'Brien, Jason L. Hirsch, Adrienne M. Orriols, Kiira Lani Amechi, Joshua Lo, Neeharika Chanda, Sarra El Hamzaoui, Jorja Kahn, Samantha V. Yap, Kyleigh E. Watson, Christina Curran, Amina Atef AbdelAlim, Neeloy Bose, Alissa Link Cilfone.

**Formal analysis:** Grace Grifferty.

**Investigation:** Grace Grifferty, Hugh Shirley, Katherine O'Brien, Jason L. Hirsch, Adrienne M. Orriols, Kiira Lani Amechi, Joshua Lo, Neeharika Chanda, Sarra El Hamzaoui, Jorja Kahn, Samantha V. Yap, Kyleigh E. Watson, Christina Curran, Amina Atef AbdelAlim, Neeloy Bose.

**Methodology:** Grace Grifferty, Hugh Shirley, Katherine O'Brien, Alissa Link Cilfone.

**Project administration:** Grace Grifferty.

**Supervision:** Grace Grifferty, Richard Wamai.

**Visualization:** Grace Grifferty.

**Writing – original draft:** Grace Grifferty, Hugh Shirley, Katherine O'Brien.

**Writing – review & editing:** Grace Grifferty, Hugh Shirley, Katherine O'Brien, Jason L. Hirsch, Adrienne M. Orriols, Kiira Lani Amechi, Joshua Lo, Neeharika Chanda, Sarra El Hamzaoui, Jorja Kahn, Samantha V. Yap, Kyleigh E. Watson, Christina Curran, Amina Atef AbdelAlim, Neeloy Bose, Richard Wamai.

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
