## [Editor Report · Decision Letter 0]

17 Oct 2022

Dear Ms. Grifferty,

Thank you very much for submitting your manuscript "The leishmaniases in Kenya: a scoping review" for consideration at PLOS Neglected Tropical Diseases. As with all papers reviewed by the journal, your manuscript was reviewed by members of the editorial board and by several independent reviewers. In light of the reviews (below this email), we would like to invite the resubmission of a significantly-revised version that takes into account the reviewers' comments. 

Editorial comments:

Manuscript's topic is certainly of interest for PLoSNTD readers. Prior to possibly sending the manuscript our for peer-review, please address the following: 

- The review is a review of the leishmaniases in Kenya. Among the 17 authors of the review, none has a Kenyan affiliation--please ensure that you have local researchers or policy makers represented, either by ensuring that dual affiliations are specified or that additional co-authors are invited.

- Abstract/Results/Discussion: the findings that there is lack of research on MCL and PKDL should be tempered, given that prevailing Leishmania species in Kenya result in no MCL/PKDL, or extremely low prevalence of MCL/PKDL, i.e. that finding is, thus, not surprising. 

- Please make sure to consult state-of-the-art reviews of the leishmaniases to ensure correct nomenclature, definitions, etc. (see: Reithinger R et al (2007) Cutaneous leishmaniasis. Lancet Infect Dis. 7: 581-596, or Burza et al. (2018) Leishmaniasis. Lancet 392: 951-970). Examples of inaccuracies are: 

* Line 86: Phlebotomus and not Phlebotomine

* Line 87 - 90: there are pathologies as well, such as disseminated cutaneous leishmaniasis. Broadly speaking you have the cutaneous leishmaniases and visceral leishmaniasis. The former includes localized cutaneous leishmaniasis, disseminated cutaneous, and mucosal leishmaniasis. Visceral leishmaniasis includes visceral leishmaniasis and post-kala-azar dermal leishmaniasis, since PKDL is a manifestations seen in patients with visceral leishmaniasis after apparent clinical cure.

* Lines 91 onwards. You are mixing up species and sub-genera. Thus, it should be <MCL is frequently seen in L. Vianna spp.> (which includes Leishmania braziliensis).

* Line 120: The sentence <Kenya, a burgeoning country in Sub-Saharan Africa, is among five East African countries that now bear the largest proportion of the global burden of leishmaniasis> cannot be correct, as there are a number of countries in LAC and Central Asia that have a much higher burden of cases) than Kenya /East Africa.

* Line 126: Please specify what you mean by <1,000 yearly>? Cases? Incidence per 1,000 population?

- Please make sure that all figures are of adequate size and resolution. The full-page Figures 1- 4 are not very legible.

- Re gaps in knowledge / research--you may want to distinguish gaps that are more pertinent to the global level (e.g., basic biology, pathobiology, immunology, co-morbidities) versus those that are East Africa/Kenya specific (e.g., vector or reservoir ecology).

We cannot make any decision about publication until we have seen the revised manuscript and your response to the reviewers' comments. Your revised manuscript is also likely to be sent to reviewers for further evaluation.

Sincerely,

Richard Reithinger

Academic Editor

Charles Jaffe

Section Editor

Editorial comments:

Manuscript's topic is certainly of interest for PLoSNTD readers. Prior to possibly sending the manuscript our for peer-review, please address the following: 

- The review is a review of the leishmaniases in Kenya. Among the 17 authors of the review, none has a Kenyan affiliation--please ensure that you have local researchers or policy makers represented, either by ensuring that dual affiliations are specified or that additional co-authors are invited.

- Abstract/Results/Discussion: the findings that there is lack of research on MCL and PKDL should be tempered, given that prevailing Leishmania species in Kenya result in no MCL/PKDL, or extremely low prevalence of MCL/PKDL, i.e. that finding is, thus, not surprising. 

- Please make sure to consult state-of-the-art reviews of the leishmaniases to ensure correct nomenclature, definitions, etc. (see: Reithinger R et al (2007) Cutaneous leishmaniasis. Lancet Infect Dis. 7: 581-596, or Burza et al. (2018) Leishmaniasis. Lancet 392: 951-970). Examples of inaccuracies are: 

* Line 86: Phlebotomus and not Phlebotomine

* Line 87 - 90: there are pathologies as well, such as disseminated cutaneous leishmaniasis. Broadly speaking you have the cutaneous leishmaniases and visceral leishmaniasis. The former includes localized cutaneous leishmaniasis, disseminated cutaneous, and mucosal leishmaniasis. Visceral leishmaniasis includes visceral leishmaniasis and post-kala-azar dermal leishmaniasis, since PKDL is a manifestations seen in patients with visceral leishmaniasis after apparent clinical cure.

* Lines 91 onwards. You are mixing up species and sub-genera. Thus, it should be <MCL is frequently seen in L. Vianna spp.> (which includes Leishmania braziliensis).

* Line 120: The sentence <Kenya, a burgeoning country in Sub-Saharan Africa, is among five East African countries that now bear the largest proportion of the global burden of leishmaniasis> cannot be correct, as there are a number of countries in LAC and Central Asia that have a much higher burden of cases) than Kenya /East Africa.

* Line 126: Please specify what you mean by <1,000 yearly>? Cases? Incidence per 1,000 population?

- Please make sure that all figures are of adequate size and resolution. The full-page Figures 1- 4 are not very legible.

- Re gaps in knowledge / research--you may want to distinguish gaps that are more pertinent to the global level (e.g., basic biology, pathobiology, immunology, co-morbidities) versus those that are East Africa/Kenya specific (e.g., vector or reservoir ecology).
---

## [Decision Letter · Decision Letter 1]

17 Mar 2023

Dear Ms. Grifferty,

Thank you very much for submitting your manuscript "The leishmaniases in Kenya: a scoping review" for consideration at PLOS Neglected Tropical Diseases. As with all papers reviewed by the journal, your manuscript was reviewed by members of the editorial board and by several independent reviewers. The reviewers appreciated the attention to an important topic. Based on the reviews, we are likely to accept this manuscript for publication, providing that you modify the manuscript according to the review recommendations. 

Sincerely,

Richard Reithinger

Academic Editor

Charles Jaffe

Section Editor

Reviewer's Responses to Questions

**Key Review Criteria Required for Acceptance?**

**Methods**

-Are the objectives of the study clearly articulated with a clear testable hypothesis stated?

-Is the study design appropriate to address the stated objectives?

-Is the population clearly described and appropriate for the hypothesis being tested?

-Is the sample size sufficient to ensure adequate power to address the hypothesis being tested?

-Were correct statistical analysis used to support conclusions?

-Are there concerns about ethical or regulatory requirements being met?

Reviewer #1: -Are the objectives of the study clearly articulated with a clear testable hypothesis stated? Yes

-Is the study design appropriate to address the stated objectives? Yes

-Is the population clearly described and appropriate for the hypothesis being tested? Yes

-Is the sample size sufficient to ensure adequate power to address the hypothesis being tested? NA

-Were correct statistical analysis used to support conclusions? Partial. See review comments

-Are there concerns about ethical or regulatory requirements being met? No

Reviewer #2: The methodology of the study are well-elaborated

**Results**

-Does the analysis presented match the analysis plan?

-Are the results clearly and completely presented?

-Are the figures (Tables, Images) of sufficient quality for clarity?

Reviewer #1: -Does the analysis presented match the analysis plan? NA

-Are the results clearly and completely presented? Partial. See additional suggestions in attached comments

-Are the figures (Tables, Images) of sufficient quality for clarity? Yes

Reviewer #2: The results section provides a clear and detailed analysis. The tables and figures are clear, but I hope the final print of figure 4 will have a better readable font.

**Conclusions**

-Are the conclusions supported by the data presented?

-Are the limitations of analysis clearly described?

-Do the authors discuss how these data can be helpful to advance our understanding of the topic under study?

-Is public health relevance addressed?

Reviewer #1: -Are the conclusions supported by the data presented? Partial. Please see attached comments

-Are the limitations of analysis clearly described? Partial. Please see attached comments

-Do the authors discuss how these data can be helpful to advance our understanding of the topic under study? Partial. Please see attached comments

-Is public health relevance addressed? Yes

Reviewer #2: The conclusions are supported by the data, and limitations are clearly described.

**Editorial and Data Presentation Modifications?**

Reviewer #1: (No Response)

Reviewer #2: (No Response)

**Summary and General Comments**

Reviewer #1: 1. Leishmaniasis is a topic that has been widely researched in Kenya, considering the volume of the publications that have me the eligibility criteria of this scoping review. The conclusions of this paper would be important to shape policy in Kenya, and as such the authors should have considered including the input of local policy makers in this kind of paper, especially in helping explain the apparent paucity of data on some aspects of Leishmaniasis research. This will give this paper a stronger contextual basis and strengthen the basis of the conclusions. The authors should be seen to have collaborated with local based policy makers and researchers for context. 

2. It would be useful if the authors would update the search strategy to include papers published in the last one year (since January 2022), given that this is a widely researched topic in Kenya. In the last few years, many publications have explored the impact of COVID-19, for example, on most neglected tropical diseases. A lot of these papers would have been published in the recent few years. It would be key to state reasons why this scoping review was not updated to the most recent data

3. The authors should explicitly explain the basis of categorizing the articles included in the review in the various ‘themes’ and subgroups, and why it is important for this paper. Ordinarily, some fields of research would attract fewer papers (e.g. reviews, since one article usually summarizes many other publications). Therefore, the metrics of comparison using proportions would be grossly misleading, since some of the fields would naturally be underrepresented.

4. In order to avoid a classification bias and to identify additional areas that have not been researched, the authors ought to have identified and contextualized (from outset) what areas of Leishmaniasis research are topical/relevant given the stage of control/elimination in Kenya. In this way, the authors would ensure that areas that are totally missing from publications are highlighted. For example, studies on modeling the burden/distribution, economic costs and risk mapping of Leishmaniasis in Kenya would be important for the elimination efforts in Kenya. This scoping review is silent on any of these types of studies.

Reviewer #2: Minor comments:

- Line 96: the specific mentioning of L. tropica for CL is quite arbitrary, as L. major and L. braziliensis are also among the most prevalence species.

- Line 96: MCL is also seen in L. aethiopica (which has potential relevance for Kenya).

- Line 111: Skin scrapes and fine needle aspiration are more frequently used methods for harvesting Leishmania than punch biopsies

-Line 118: Pentamidine is not recommended for treatment of VL, due to its toxicity in therapeutic regimens. In East Africa paromomycin is used as first line treatment in combination with SSG, and miltefosine is also used as second line (in combination with liposomal amphotericin.

- Line 120: topical paromomycin is not commercially available. Pentavalent antimonials are the mainstay in Old World CL, and since last year PAHO has recommended miltefosine as first line treatment of New World CL.

- Line 305: GBD needs to be spelled out (Global Burden if Disease?)

- Line 319: authors mention a prevalence (should this not be incidence?) of 2-5%. However, this is just an estimation. A recent clinical trial in East Africa (Musa et al, Clin.Inf.Dis. 2022) found no PKDL in Kenya in a 6 months follow up after treatment of VL. Because the PKDL incidence is not known, and because of the public health implications of PKDL, I would recommend authors to make a recommendation for PKDL incidence studies.

- Line 324: Rightfully HIV is mentioned as an important co-infection. However, TB is another important co-infection, due to the relatively high prevalence of pulmonary and extra-pulmonary TB in VL patients (re-activation of latent TB infection due to VL-caused immune-suppression), and the fact that undiagnosed TB is a main reason for VL treatment failure.

PLOS authors have the option to publish the peer review history of their article (what does this mean?). If published, this will include your full peer review and any attached files.

Reviewer #1: No

Reviewer #2: No

Figure Files:

Data Requirements:

Reproducibility:

References

---

## [Editor Report · Decision Letter 2]

7 May 2023

Dear Ms. Grifferty,

We are pleased to inform you that your manuscript 'The leishmaniases in Kenya: a scoping review' has been provisionally accepted for publication in PLOS Neglected Tropical Diseases.

Best regards,

Richard Reithinger

Academic Editor

Charles Jaffe

Section Editor

Thank you for addressing all if the reviewers' comments and queries.

---

## [Editor Report · Acceptance letter]

26 May 2023

Dear Ms. Grifferty,

We are delighted to inform you that your manuscript, "The leishmaniases in Kenya: a scoping review," has been formally accepted for publication in PLOS Neglected Tropical Diseases.

Best regards,

Shaden Kamhawi

co-Editor-in-Chief

Paul Brindley

co-Editor-in-Chief
